# A Metabolism-Related Gene Prognostic Index for Prediction of Response to Immunotherapy in Lung Adenocarcinoma

**DOI:** 10.3390/ijms232012143

**Published:** 2022-10-12

**Authors:** Bo Tang, Lanlin Hu, Tao Jiang, Yunchang Li, Huasheng Xu, Hang Zhou, Mei Lan, Ke Xu, Jun Yin, Chunxia Su, Caicun Zhou, Chuan Xu

**Affiliations:** 1Integrative Cancer Center and Cancer Clinical Research Center, Sichuan Cancer Hospital and Institute, Sichuan Cancer Center, School of Medicine, University of Electronic Science and Technology of China, Chengdu 610042, China; 2Department of Medical Oncology, Shanghai Pulmonary Hospital, Tongji University School of Medicine, No. 507, Zhengmin Road, Shanghai 200433, China; 3Collaborative Innovation Centre of Regenerative Medicine and Medical BioResource Development and Application, Guangxi Medical University, Nanning 530021, China

**Keywords:** prognostic index, metabolism-related genes, lung adenocarcinoma, immunotherapy

## Abstract

Immunotherapy, such as immune checkpoint inhibitors (ICIs), is a validated strategy for treating lung adenocarcinoma (LUAD) patients. One of the main challenges in ICIs treatment is the lack of efficient biomarkers for predicting response or resistance. Metabolic reprogramming has been proven to remodel the tumor microenvironment, altering the response to ICIs. We constructed a prognostic model as metabolism-related gene (MRG) of four genes by using weighted gene co-expression network analysis (WGCNA), the nonnegative matrix factorization (NMF), and Cox regression analysis of a LUAD dataset (*n* = 500) from The Cancer Genome Atlas (TCGA), which was validated with three Gene Expression Omnibus (GEO) datasets (*n *= 442, *n *= 226 and *n* = 127). The MRG was constructed based on *BIRC5*, *PLK1*, *CDKN3*, and *CYP4B1* genes. MRG-high patients had a worse survival probability than MRG-low patients. Furthermore, the MRG-high subgroup was more associated with cell cycle-related pathways; high infiltration of activated memory CD4^+^T cells, M0 macrophages, and neutrophils; and showed better response to ICIs. Contrarily, the MRG-low subgroup was associated with fatty acid metabolism, high infiltration of dendric cells, and resting mast cells, and showed poor response to ICIs. MRG is a promising prognostic index for predicting survival and response to ICIs and other therapeutic agents in LUAD, which might provide insights on strategies with ICIs alone or combined with other agents.

## 1. Introduction

Lung cancer is one of the common malignancies with high mortality worldwide [1] and is histologically classified into non-small cell lung cancer (NSCLC) and small cell lung cancer (SCLC). NSCLC can be further classified into lung squamous carcinoma, lung adenocarcinoma (LUAD), and large lung carcinoma, among which LUAD is the most common subtype with high molecular heterogeneity [2,3].

Therapeutic strategies for LUAD include surgical resection, radiotherapy, chemotherapy, targeted therapy, and immunotherapy [4]. Although the diagnosis of LUAD has improved in recent years, the recurrence and the survival rates remain unfavorable [5]. With the approval of immune checkpoint inhibitors (ICIs) in LUAD treatment, the therapeutic paradigm in LUAD has been altered [6]. Although LUAD patients could benefit from ICIs treatment, there remains a proportion of LUAD patients showing no response [7]. The most commonly used biomarkers for ICIs treatment include expression of programmed death-ligand 1 (PD-L1), microsatellite instability (MSI), and tumor mutation burden (TMB), all of which have been inconsistent in predicting ICIs efficacy in many types of cancer, including LUAD [8,9,10]. Accordingly, it would be helpful to develop more reliable and effective biomarkers to predict the response to ICIs treatment.

Metabolic reprogramming is one of the hallmarks of cancer [11]. Studies have investigated metabolism in cancers and tumor microenvironment (TME) extensively over the past decades, which indicates the promising potential of metabolic targets to modulate anti-cancer immunity [12]. The alteration of metabolism in cancer cells might result in the accumulation of metabolites in TME or competition of nutrients between cancer cells and stromal cells in TME, which leads to immunosuppressive TME and, subsequently, poor prognosis in patients. Therefore, metabolism-related genes (MRGs) as prognostic markers for LUAD and biomarkers for ICIs treatment in LUAD are promising. 

In this study, based on MRGs, we constructed a signature prognostic index from The Cancer Genome Atlas (TCGA) database, which was further validated by the data from the Gene Expression Omnibus (GEO) database. Additionally, we investigated the molecular and cellular signature differences between patients in the MRG-low and the MRG-high subgroups and predicted their prognostic potential in ICIs. Moreover, the sensitivity to different therapeutic agents in both subgroups was analyzed. The results indicate that MRG is a promising prognostic index for predicting prognosis and a biomarker for predicting patients’ responses to ICIs.

## 2. Results

### 2.1. Identification of Metabolism-Related Hub Genes

In total, 8190 DEGs were identified by analyzing RNA-seq data from the TCGA database (535 tumors vs. 59 normal samples) (*p* < 0.05, |log2FC| > 1), containing 6316 upregulated and 1874 downregulated genes in the tumor compared with the normal samples (Appendix A). By intersecting these DEGs with the lists of MRGs obtained from the GeneCards, 1441 DEMRGs were identified, composed of 966 upregulated genes and 475 downregulated genes between the tumor and the normal samples (Appendix A). The functional enrichment analysis showed that 1,441 DEMRGs were markedly associated with 3007 GO terms and 43 KEGG pathways, and the top 10 GO terms and KEGG pathways are shown in Appendix A, respectively.

To obtain the metabolism-related hub genes, the 1441 DEMRGs were analyzed by WGCNA. Based on the scale-free network, the optimal soft threshold is 3 (Appendix A). Then, eight modules were identified based on the average linkage hierarchical clustering and the optimal soft-thresholding power (Appendix A). The DEMRGs were assigned to eight modules. Based on Pierson’s correlation coefficient between a module and sample feature for each module, the genes in the turquoise module were significantly negatively correlated with LUAD, and the genes in the blue module markedly positively correlated with LUAD. A total of 1089 genes in the turquoise and the blue module were processed for functional enrichment analysis, and the top 10 enriched GO terms and KEGG pathways are shown in Appendix A, respectively. 

The univariate Cox analysis was performed on the 1089 DEMRGs, and 72 genes closely related to the prognosis of LUAD were identified and further processed for NMF analysis (Appendix A; *p* < 0.001). According to the NMF algorithm, the 72 genes were optimally fractionated into two subtypes, C1 and C2 (Appendix A). The patients in the C2 subtype showed poorer overall survival (OS) (Appendix A) and progression-free survival (PFS) (Appendix A) than those in the C1 subtype (log-rank *p* < 0.05). CIBERSORT helps profile tumor-infiltrating immune cells from bulk tissue gene expression data [13]. The CIBERSORT results indicated that CD8^+^ T cells, activated memory CD4^+^ T cells, and M1 macrophages in the patients from the C2 subtype were significantly higher than those in the patients from the C1 subtype (Appendix A). Accordingly, we speculated that the C2 subtype might respond better to immunotherapy. We analyzed the DEMRGs between C1 and C2 and obtained 97 DEMRGs between two subgroups. The top 20 DEMRGs are shown in Appendix A. 

### 2.2. Survival Analysis of the Different MRG Subgroups

To further identify the independent prognostic genes, we conducted the univariate Cox regression analysis for survival probability and determined a list of 15 DEMRGs closely related to the prognosis of LUAD (Figure 1A,B and Appendix A).

To determine the independent prognostic genes, the multivariate Cox regression analysis for overall survival was performed among the 15 DEMRGs [6]. We obtained four genes (*BIRC5*, *PLK1*, *CDKN3*, and *CYP4B1*) with a minimal AIC value of 1919.42. Then, a prognostic index for all cancer was constructed by the formula MRG = expression level of *BIRC5* × (−0.4126) + expression level of *PLK1* × 0.39986 + expression level of *CDKN3* × 0.2651 + expression level of *CYP4B1* × (−0.0955). 

The clinicopathological characteristics of patients with LUAD in TCGA are shown in Appendix A. Using the median MRG as the cut-off value, the samples with MRG greater than the median value are assigned to the MRG-high subgroup, whereas those with less than the median value are assigned to the MRG-low subgroup (Appendix A). The expression pattern of four genes was also presented in the heatmap (Appendix A).

The univariate Cox regression analysis showed that Stage, T, N, and MRG were significantly associated with the prognosis of LUAD (Figure 1C). After adjusting other clinicopathologic factors the MRG score was identified as an independent prognostic factor by multivariate Cox regression analysis (Figure 1D). Survival analysis showed that the MRG-low subgroup had a better OS than the MRG-high subgroup (*p* < 0.001, log-rank test; Figure 1E). The LUAD GEO datasets (GSE68465 (*n* = 442), GSE31210 (*n* = 226) and GSE50081 (*n* = 127)) were used to validate the role of MRG (Appendix A). Survival analysis showed that the MRG-low subgroup had a significantly better prognosis than the MRG-high subgroup, consistent with the result of the TCGA dataset (Appendix A, *p* = 0.006, *p* = 0.014 and *p* = 0.045, log-rank test). The prognostic performance of the MRG was measured with the area under the curves (AUCs). The AUCs of MRG in the TCGA cohort for one-, two-, and three-year survival were 0.659, 0.669, and 0.674, respectively (Figure 1F); the AUCs of MRG in GEO cohorts also shown in Appendix A.

### 2.3. Molecular Characteristics of Different MRG Subgroups

To explore the different molecular characteristics between the two MRG subgroups, the GSEA analysis of TCGA and GEO cohorts with annotations of KEGG was performed (Appendix A). The MRG-high subgroup was enriched in cell cycle, DNA replication, chemokine signaling pathway, and so forth. (Figure 2A–D; *p* < 0.05, FDR < 0.25). The MRG-low subgroup was enriched in metabolism-related pathways. (Figure 2E–H; *p* < 0.05, FDR < 0.25). When further analyzing the difference in metabolic processes between the two subgroups, we found that genes involved in glycosaminoglycan biosynthesis, riboflavin metabolism, pentose phosphate pathway, one-carbon metabolism, and pyrimidine metabolism were enriched in the MRG-high subgroup. In contrast, genes involved in ascorbate and aldarate metabolism, fatty acid metabolism, valine, leucine and isoleucine degradation, histidine metabolism, butanoate metabolism, primary bile acid biosynthesis, and histidine metabolism were more abundant in the MRG-low subgroup (Appendix A; Figure 3A).

The gene mutations were also analyzed. Maftools analysis showed that missense mutation was the most common mutation type, followed by nonsense and multi-hit in both subgroups. The top 20 genes with the highest mutation rates (mutation rate ≥ 10%) in both MRG subgroups are shown in Figure 3B,C. In brief, the mutation rates are higher in the MRG-high subgroup than those in the MRG-low subgroup (Figure 3D, *p* = 9.5 × 10^–14^). When analyzing the Pearson’s correlation between MRG score and tumor mutation burden (TMB) of all samples from the TCGA database, the MRG score was positively correlated with TMB (R = 0.4, *p* < 2.2 × 10^–16^) (Figure 3E). 

### 2.4. Immune Characteristics of Different MRG Subgroups

To understand the immune signatures in both subgroups, we analyzed the expression of immunomodulators, including MHC molecules, immune checkpoint molecules, and cytokines. As shown in Figure 4A, Appendix A, the expression of transporter associated with antigen processing (TAP) genes (TAP1 and TAP2) were higher in the MRG-high subgroup, whereas the expressions of MHC-II molecules are higher in the MRG-low subgroup. The expression of inhibitory immune checkpoint molecules, such as PD-L1, PD-L2, and LAG3, tends to be higher in the MRG-high subgroup (Figure 4B, Appendix A). Noteworthy, most cytokines with anti-tumor activity, such as IFN-γ and CXCL10, have significantly higher expression in the MRG-high subgroup (Figure 4C, Appendix A).

CIBERSORT results showed that the MRG-low subgroup had a higher abundance of tumor-promoting immune cells, including monocytes and resting mast cells, and other immune cells, such as resting CD4^+^ T memory cells and resting DCs. In the MRG-high subgroup, neutrophils and anti-tumor immune cells, such as CD8^+^ T cells, activated memory CD4^+^ T cells, and M1 macrophages, were more abundant (Figure 5A–D). 

When investigating the molecular and immune-related function of two subgroups by using ssGSEA, we found more abundance of inflammation-promoting features, NK-cells, MHC-class-I, and Th1-cells in the MRG-high subgroup, while aDCs, B-cells, resting Dendritic DCs, HLA, iDCs, mast-cells, neutrophils, pDCs, T-helper-cells, TIL and Type-II-IFN-Response were more abundant in the MRG-low subgroup (Appendix A).

Thorsson et al. [14] analyzed over 10,000 tumors and proposed six immune subtypes to define a distinct immune response. Accordingly, we further clustered LUAD samples into five subtypes, C1 (wound healing), C2 (IFN-γ dominant), C3 (inflammatory), C4 (lymphocyte depleted), and C6 (TGF-β dominant). As shown in Appendix A, two subgroups were almost evenly distributed in C2, C4, and C6, whereas there were more MRG-high patients in C1 and more MRG-low patients in C3 (*p* = 0.001, X^2^ test). C1 is characterized by high angiogenic gene expression, high proliferation ratio, and Th2 biased acquired immune infiltration. As for C3, the proliferation of tumor cells cannot be effectively controlled.

### 2.5. The Benefit of Therapeutic Agents in Different MRG Subgroups

TIDE (Tumor Immune Dysfunction and Exclusion) is a computational framework mimicking tumor immune escape mechanisms established by Peng Jiang et al. [15] and can predict the response to ICIs based on treatment-initiation data from more than 33,000 samples in 189 studies. We used TIDE to assess the clinical efficacy of ICIs in two subgroups and found that patients in the MRG-high subgroup had lower TIDE score (Figure 6A), indicating that those patients might respond better to ICIs. The datasets of LUAD (GSE126044 (*n* = 16) and GSE93157 (*n* = 12)), urothelial carcinoma (IMvigor210 (*n* = 298)), and melanoma (GSE78220 (*n* = 28) and PRJEB23709 (*n* = 54)) were processed for evaluating MRG. As shown in Figure 6B, the LUAD patients with a higher MRG score were more likely to respond to ICI (Wilcoxon test, *p* = 0.011). The clinicopathological characteristics of LUAD patients with response status are shown in Appendix A. In the cohort of urothelial carcinoma, patients who responded to ICI better also showed higher MRG score (Appendix A). Survival analysis results showed that MRG-low patients with urothelial carcinoma have poor prognosis than MRG-high patients (Appendix A, *p* = 0.063).

We analyzed the sensitivity to some targeted and chemotherapeutic agents commonly adopted in the clinical treatment of LUAD using the OncoPredict model. The results showed that patients in the MRG-high subgroup might be more sensitive to chemotherapeutic agents, including docetaxel, paclitaxel, and vinorelbine. (Figure 6C–F). 

In brief, MRG can be a promising prognostic index in predicting survival and response to ICIs and other drugs, but it needs more data from the real world to verify.

## 3. Discussion

Cancer is a dynamic and heterogeneous disease with distinct molecular signatures [16,17]. Numerous gene signatures have been established to predict the prognosis of cancer. Current biomarkers applicated for predicting diagnosis encompass PD-L1 expression, TMB, and MSI, but the efficacy of prediction varies in different cohorts of patients [10]. Since Otto Warburg discovered that tumor cells depend on aerobic glycolysis other than mitochondrial oxidative phosphorylation in 1924, numerous investigations have uncovered the heterogeneity of cancer metabolism, which might become the potential therapeutic target in cancer treatment [18,19,20]. In this study, we established a prognostic index, MRG, of four MRGs by analyzing public data from TCGA and validated it by the data from GEO. Our gene signature, MRG, effectively stratified patient survival outcomes in the LUAD cohort. 

The four genes included in MRG have a distinct pattern of expression in two subgroups. Three genes, Baculoviral IAP repeat containing 5 (*BIRC5),* Polo like kinase 1 (*PLK1)*, and Cyclin dependent kinase inhibitor 3 (*CDKN3)*, have a high level of expression in the MRG-high subgroup, whereas Cytochrome P450 Family 4 Subfamily B Member 1 (*CYP4B1)* has a higher level of expression in the MRG-low subgroup. To understand the cause of the difference in molecular and immune-related signatures between two subgroups, we investigated the functions of four genes. CYP4B1 is a member of the cytochrome P450 superfamily of monooxygenases that catalyze drug metabolism and synthesis of cholesterol, steroids, and other lipids [21]. Studies have uncovered the role of CYP4B1 in the oxidation of fatty acids, activation of procarcinogens and neovascularization, and the promising potential as a target in cancer treatment [22,23]. PLK1 is a Serine/Threonine protein kinase of the CDC5/Polo subfamily and a critical cell cycle regulator frequently overexpressed in cancers [24]. Recent studies have reported the regulatory function of PLK1 on the pentose phosphate pathway [25] and insulin signaling [26]. BIRC5 is a member of the inhibitor of apoptosis (IAP) gene family and a negative regulatory protein with dual roles in promoting cell proliferation and preventing apoptosis [27,28]. It has been demonstrated that BIRC5 could induce autophagy through stabilizing autophagy related 7 (ATG7) protein and promoting the expression of ATG12-ATG5 conjugate [29]. BIRC5 has also been reported to enhance aerobic glycolysis in neuroblastoma [30,31]. CDKN3, a member of the dual specificity protein phosphatase family, could avert the activation of cyclin-dependent kinase 2 (CDK2) kinase through dephosphorylating CDK2 kinases [31]. The association between CDKN3 and poor prognosis has been reported in LUAD [32]. All three genes, *PLK1*, *BIRC5*, and *CDKN3*, have been described to play a vital role in cell cycle and mitosis, which corresponds to the GSEA analysis results of the MRG-high subgroup (Figure 2A). In the MRG-high subgroup, genes associated with one-carbon metabolism by folate were also more abundant. Plentiful evidence has revealed the vital role of folate-mediated one-carbon metabolism in the growth and proliferation of cancer cells [33]. PLK1 has recently been reported to phosphorylate methylenetetrahydrofolate reductase (MTHFR), an essential enzyme in one-carbon metabolism mediated by folate [34]. Studies mentioned above might elucidate the enrichment of one-carbon metabolism in the MRG-high subgroup. Nevertheless, genes enriched in the MRG-low subgroup were involved in different types of metabolism such as the metabolism of branched-chain amino acids (BCAAs) and fatty acids metabolism. BCAAs are one class of amino acids having an aliphatic sidechain with a branch, including valine, leucine, and isoleucine. They have been reported to be utilized as the nitrogen sauce for nonessential amino acid and nucleotide synthesis and the energy sauce for ATP production [35,36,37]. The disorder of fatty acid metabolism has also been demonstrated in tumorigenesis and therapeutic resistance, which is well reviewed [38]. Despite the direct or indirect association of CYP4B1, PLK1, and BIRC5 in the metabolic signaling pathway, further investigation might be needed to unveil the relationship between CDKN3 and other genes and metabolisms in the MRG subgroup. 

Since 2015, several ICIs have been approved by US Food and Drug Administration (FDA) for the treatment of advanced-stage NSCLC [39]. Although ICIs have greatly improved the prognosis of LUAD, studies have found that some patients still showed disease progression [7]. Previous investigations have revealed that the response to ICIs could be affected by many risk factors, such as gene mutations, pro-inflammatory cytokines, etc. [40]. We analyzed the mutation status in two MRG subgroups and found that the overall TMB was higher in the MRG-high subgroup. However, the mutation type of common mutations, such as EGFR and KRAS, in two MRG subgroups showed no significant difference. Therefore, we assumed that gene mutation is not the leading cause of different responses to ICIs in two subgroups. The metabolic reprogramming comprehends alterations in cancer and stromal cells in TME-regulated cell-intrinsic factors and metabolites in a particular microenvironment. Growing evidence has demonstrated that TME could be shaped by intrinsic cancer cell metabolism, the interplay between cancer cells and stromal cells in TME, tissue specificity and heterogeneity, and metabolic homeostasis in the whole body [41,42,43]. Therefore, we speculated that the different metabolic signatures in two MRG subgroups lead to contrary immune phenotypes, which subsequently bring about divergent responses to immunotherapy. Accordingly, we analyzed the expression of different immunomodulators and the distribution of infiltrating immune cells in two MRG subgroups. Currently, the ICIs in clinical application employ antibodies to block receptor–ligand interactions. In many clinical applications, the expression level of immune checkpoint molecules, especially PD-1/-L1, is used as the criteria for treatment. Hence, the high expressing level of immune checkpoint molecules might explain why most patients with better responses to ICIs had higher MRG score (Figure 4A, Appendix A). The antigen presentation is a complex process important for eliciting an effective anti-tumor response. The mechanisms involved in immune evasion include alteration of antigen expression, HLA-I surface levels, changes in antigen processing and presentation in tumor cells [44]. We found that antigen processing genes (TAP1/2) are higher in patients from the MRG-high subgroup than those from the MRG-low subgroup. TAPs promote MHC I folding and the loading of peptides into the proper MHC I peptide binding groove, which is essential in MHC I antigen presentation. The higher expression level of TAPs in patients from MRG-high subgroup might indicate that more antigens presented on the surface of tumor cells could be recognized by effector T cells. Neutrophils have been demonstrated to promote tumor development and anti-tumor activities by regulating angiogenesis and cell detachment [45]. Neutrophil infiltration has also been reported to predict the resistance to ICIs [46]. Contrarily, our results showed more neutrophil infiltration in the MRG-high subgroup. Additionally, we identified immune subtypes in LUAD patients from the TCGA cohort. Thorsson et al. have identified six immune subtypes, wound healing, IFN-γ dominant, inflammatory, lymphocyte depleted, immunologically quiet, and TGF-β domain, by performing an immunogenomic analysis of over 10,000 tumors comprising 33 cancer types. Different subtypes are characterized by differences in macrophage or lymphocyte signatures, Th1:Th2 cell ratio, intratumoral heterogeneity, immunomodulatory gene expression, etc. [14]. As shown in Appendix A, patients from MRG-low and -high subgroups were distributed almost evenly in C1, C2, C3, C4, and C6 immune subtypes, and no LUAD patients could be categorized into the C5 immune subtype. C5 subtype (immunological quiet) showed higher population of M2 macrophage, the lowest lymphocyte, and the highest macrophage responses, which is consistent with our result that no difference in M2 macrophage infiltration was detected in MRG subgroups. Although LUAD patients in C1 (wound healing) subtypes are more likely to be categorized into MRG-high subgroup, and LUAD patients in C3 (inflammatory) subtypes are more likely to have low MRG score, and no direct connection could be revealed between MRG subgroups and immune subtypes. Overall, the immunological characteristics could not be fully defined. More experiments are required to further understand the population of immune cells, such as neutrophiles, macrophages, etc., in MRG subgroups and its association with ICI response. 

TIDE is an algorithm developed to predict patients’ responses to ICIs by Peng et al. It was reported that patients with higher TIDE score are likely to have more significant immune evasion and less likely to benefit from ICI treatment [15]. When comparing our MRG with the TIDE in predicting response to ICIs in cohorts of LUAD patients from the TCGA database, the MRG-high subgroup had a lower TIDE score, which indicates a better response to ICIs. Since chemotherapy and targeted therapy remain the main strategies carried out alone or combined with other strategies in LUAD treatment, we analyzed the prognostic value of MRG in predicting response to some agents in the clinical application of LUAD treatment. The MRG-high subgroup showed sensitivity to chemotherapeutic drugs targeting mitosis and nucleic metabolism. GSEA analysis has shown more genes enriched in cell cycle and DNA replication pathways, which might explain the better sensitivity to docetaxel, paclitaxel, and vinorelbine. The accuracy needs more preclinical and clinical data to verify its practice. 

When we validated the prognostic value of MRG in the other three datasets encompassing urothelial carcinoma and melanoma, only two cohorts of melanoma (GSE78220 and PRJEB23709) showed consistent results with no significance (Appendix A, *p* = 0.090 and *p* = 0.257, respectively). We assumed that the sample size of these three cohorts was comparatively small. As for the cohort of urothelial carcinoma, the MRG-low subgroup showed a worse survival probability than the MRG-high subgroup with no significance, which might owe to distinct metabolic patterns caused by tissue specificity and the different microenvironment between urothelial carcinoma and LUAD. Moreover, the response to ICIs showed a correlation with MRG score in all cohorts, including patients with LUAD, urothelial carcinoma, and melanoma, even though the *p* value is more than 0.05 in melanoma cohorts. The sample size of LUAD and melanoma is relatively small, and more clinical data are required for further validation before application in clinics.

## 4. Materials and Methods

### 4.1. Data Source and Preprocessing

The RNA-seq data and clinicopathologic information ((including T stage, Survival time, Age, Gender, T stage, N stage, and Status) of LUAD samples (*n* = 500) were downloaded from TCGA Genomic Data Commons Data Portal. Additionally, the RNA-seq data and clinical information of LUAD samples (GSE68465 (*n* = 442), GSE31210 (*n* = 226) and GSE50081 (*n* = 127)) were downloaded from the GEO database. The clinical data from patients treated with ICIs were downloaded from the Tumor Immune Dysfunction and Exclusion (TIDE) website (GSE78220, GSE93157, IMvigor210, PRJEB23709, GSE126044). 

### 4.2. Acquisition of MRGs

MRGs were downloaded from the GeneCards database. Metabolism was used as the keyword to search for genes in the search term, and genes with a relevance score ≥2 were selected as MRGs.

### 4.3. Differentially Expressing Genes (DEGs) Analysis

DEGs between tumor samples and normal samples from the RNA-seq data of LUAD samples obtained from TCGA were identified by differential expression analysis using the “limma” package in R (*p* < 0.05, |log2FC| > 1). MRGs were intersected with DEGs to obtain differentially expressing metabolism-related genes (DEMRGs) that were used for further analysis.

### 4.4. Gene Ontology Enrichment and Kyoto Encyclopedia of Genes and Genomes Analyses 

To explore in-depth the possible biological processes (BP), cellular components (CC), molecular functions (MF), and signaling pathways of the common DEGs, Gene Ontology (GO) enrichment and Kyoto Encyclopedia of Genes and Genomes (KEGG) analysis were performed by utilizing the “clusterProfiler” package in R with a statistical threshold of *p* < 0.05 as described previously [47].

### 4.5. Identification of Metabolism-Related Hub Genes and LUAD Subgroups

WGCNA was performed to recognize co-expressed hub gene modules. The threshold for the determination of weighted adjacency matrix was fixed at a soft threshold of β = 3 and a scale-free R^2^ > 0.85. We identified eight modules by setting the merging threshold. Based on the genes of significantly related modules (the turquoise and blue module), we used nonnegative matrix factorization (NMF) to perform the molecular subgroup [48]. First, metabolism-related hub gene expression profiles were extracted from the TCGA-LUAD cohort, and univariate Cox analysis was performed to obtain prognostic genes related (*p* < 0.001). A total of 500 LUAD samples were divided into two distinct molecular subgroups with significantly different prognoses (C1 and C2) based on metabolic prognosis-related genes. The NMF method selects the standard “Brunet” and performs 100 iterations. The cluster number K was set to 2–10, the average contour width of the common member matrix was determined by the R package “NMF,” and the minimum membership of each subclass was set to 10.

### 4.6. Construction and Validation of the MRG

Using the “limma” package in R (*p* < 0.05, |log2FC| > 1) to identify DEGs between the C1 and C2 subtypes. The top 20 differentially expressed genes were selected to construct a prognostic model. First, 15 genes associated with prognosis were found using univariate cox analysis (*p* < 0.001). Then, among the 15 metabolism-related genes, the genes that affect the survival of LUAD patients were identified by multivariate Cox regression analysis. The MRG score for each patient is obtained by multiplying the model gene expression values of each patient by the weights of these genes in the Cox model and then adding them together. The prognostic power of the MRG model was assessed by Kaplan–Meier (K-M) survival curves and log-rank tests for the TCGA-LUAD and GEO-LUAD (GSE68465, GSE31210 and GSE50081) cohorts. To verify the independent prognostic value of MRG, we performed univariate and multivariate Cox regression analyses.

### 4.7. Comprehensive Analysis of Molecular and Metabolism Characteristics

In the signaling function analysis, GSEA and GSVA were performed to determine the relevant pathways and molecular mechanisms in the high and low MRG subgroups in the TCGA and GEO cohort based on the hallmark gene set (c2.cp.kegg.v7.4.symbols.gmt) derived from the Molecular Signatures Database (MSigDB)database. In the gene mutation analysis, information on genetic alterations was obtained from the TCGA database, and the quantity and quality of gene mutations were analyzed in two MRG subgroups by using the Maftools package of R [49]. Correlation analyses were performed between MRG score and TMB. 

To identify the metabolism characteristics of all LUAD samples, their expression data were imported into CIBERSORT and simulated 1000 times to estimate the relative proportion of 22 types of immune cells [50]. Then, we compared the relative proportions of 22 types of immune cells and clinicopathologic factors between the two MRG subgroups, and the results are presented in a landscape map. To further define the immune cell and molecular function between the MRG subgroups, we performed the single simple gene set enrichment analysis (ssGSEA) on certain gene signatures and compared the score between two MRG subgroups [51].

### 4.8. The Benefit of Therapeutic Agents in Different MRG Subgroups

To predict the half-maximal inhibitory concentration (IC_50_) of chemotherapy drugs in the high-MRG and low-MRG subgroups of LUAD patients and to infer the sensitivity of the different patients, we used the “oncoPredict” R package. By constructing the ridge regression model based on Genomics of Drug Sensitivity in Cancer (GDSC) (www.cancerrxgene.org/, accessed on 15 July 2021) cell line expression spectrum and TCGA gene expression profiles, the package could apply oncoPredict algorithm to predict drug IC_50_ [52].

To explore the prognostic value of MRG in patients after immunotherapy, we used the TIDE to assess the response to immunotherapy in different MRG subgroups. At the same time, we performed survival analysis and immune response analysis in multiple cohorts treated with anti-PD-L1 [53].

### 4.9. Statistical Analysis

All statistical analyses were conducted in R software 4.1.0. An independent t test was performed to compare continuous variables between two subgroups. Categorical data were tested using the X^2^ test, *p* < 0.05. TIDE score between subgroups was compared by the Wilcoxon test. Univariate survival analysis was performed by K-M survival analysis with the log-rank test. Multivariate survival analysis was performed using the Cox regression model. *p* < 0.05 was considered significant.

## 5. Conclusions

In summary, we established a metabolism-related gene signature named by MRG that showed good prognostic value in predicting survival probability and response to ICIs in LUAD. Moreover, the MRG also showed good performance when applied to predict chemotherapeutic response in LUAD, which might give an indication of combinational strategies in ICI treatment. However, more data from the real world and basic research are demanded for verification and validation prior to application in clinics.

## Figures and Tables

**Figure 1 ijms-23-12143-f001:**
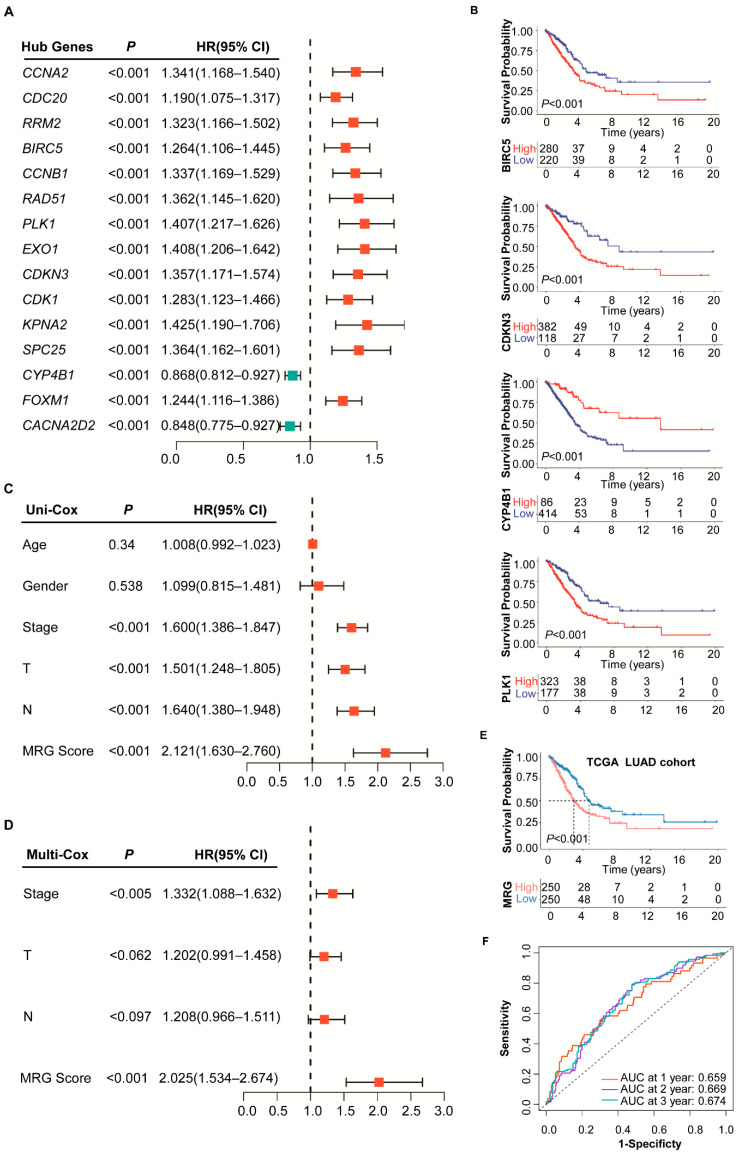
Prognostic analysis of different MRG subgroups. (**A**) The univariate Cox regression analysis of 15 MRGs. (**B**) K-M survival analysis of four metabolism-related genes in MRG index. (**C**) The univariate Cox regression analysis of clinicopathological factors and the MRG score. (**D**) The multivariate Cox regression analysis of factors significant in the univariate Cox regression analysis (*p* < 0.05). (**E**) K-M survival analysis of MRG subgroups in TCGA cohort. (**F**) ROC analysis of MRG on prediction of 1-, 2-, and 3-year survival in TCGA cohort.

**Figure 2 ijms-23-12143-f002:**
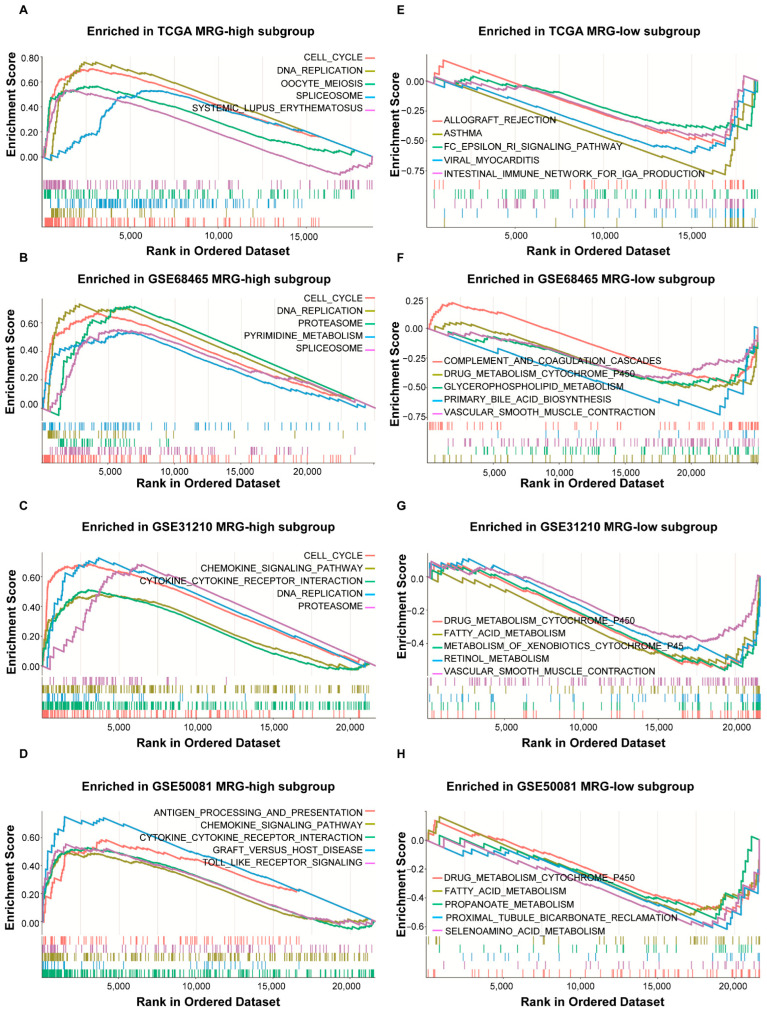
Gene Sets Enrichment Analysis of different MRG subgroups. (**A**) Gene Sets Enrichment Analysis (GSEA) of KEGG pathways in the TCGA MRG-high subgroups. (**B**) GSEA of KEGG pathways in the GSE68465 MRG-high subgroups. (**C**) GSEA of KEGG pathways in the GSE31210 MRG-high subgroups. (**D**) GSEA of KEGG pathways in the GSE50081 MRG-high subgroups. (**E**) GSEA of KEGG pathways in the TCGA MRG-low subgroups. (**F**) GSEA of KEGG pathways in the GSE68465 MRG-low subgroups. (**G**) GSEA of KEGG pathways in the GSE31210 MRG-low subgroups. (**H**) GSEA of KEGG pathways in the GSE50081 MRG-low subgroups; *p* < 0.05.

**Figure 3 ijms-23-12143-f003:**
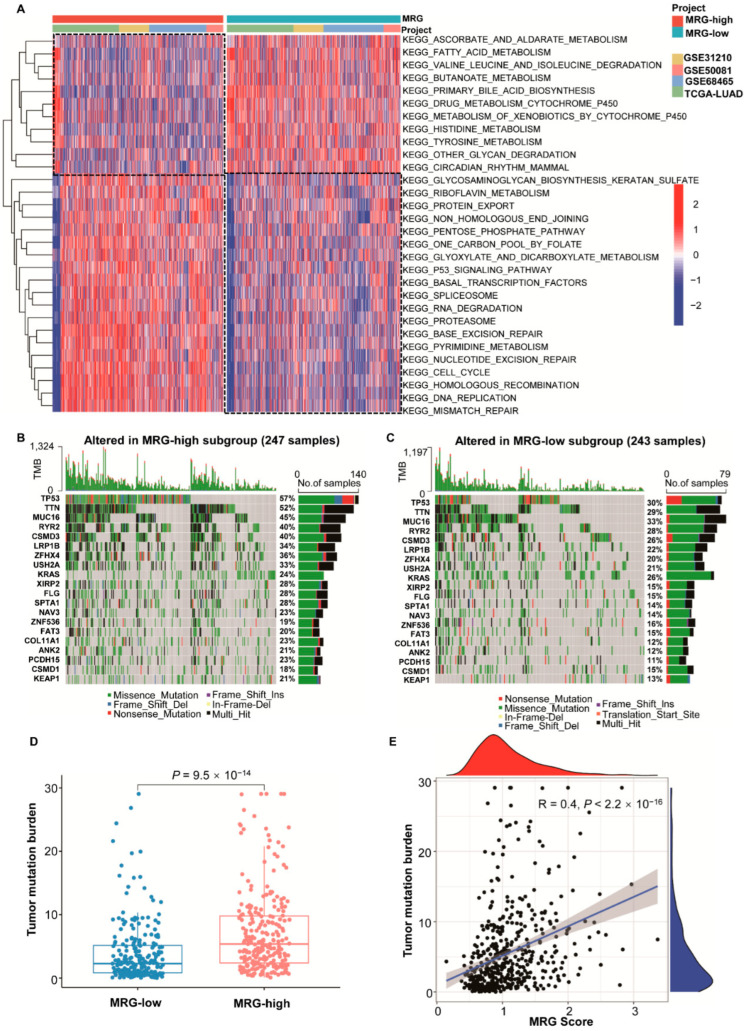
Molecular characteristics of different MRG subgroups. (**A**) Heatmap of Gene Set Variation Analysis (GSVA) of KEGG pathways in individual LUAD patients from TCGA and GEO database. (**B**) Significantly mutated genes in different MRG-high subgroups. Mutated genes (rows, top 20) are ordered by mutation rate; samples (columns) are arranged to emphasize mutual exclusivity among mutations. The mutation percentage is shown on the right, and the overall number of mutations is shown on the top. The color coding indicates the mutation type. (**C**) Significantly mutated genes in different MRG-low subgroups. Mutated genes (rows, top 20) are ordered by mutation rate; samples (columns) are arranged to emphasize mutual exclusivity among mutations. The mutation percentage is shown on the right, and the overall number of mutations is shown on the top. The color coding indicates the mutation type; (**D**) Overall tumor mutation burden (TMB) status in different subgroups. *p* value is shown in the figures. (**E**) Pearson’s correlation of TMB and MRG score. *p* value and R value are shown in the figures.

**Figure 4 ijms-23-12143-f004:**
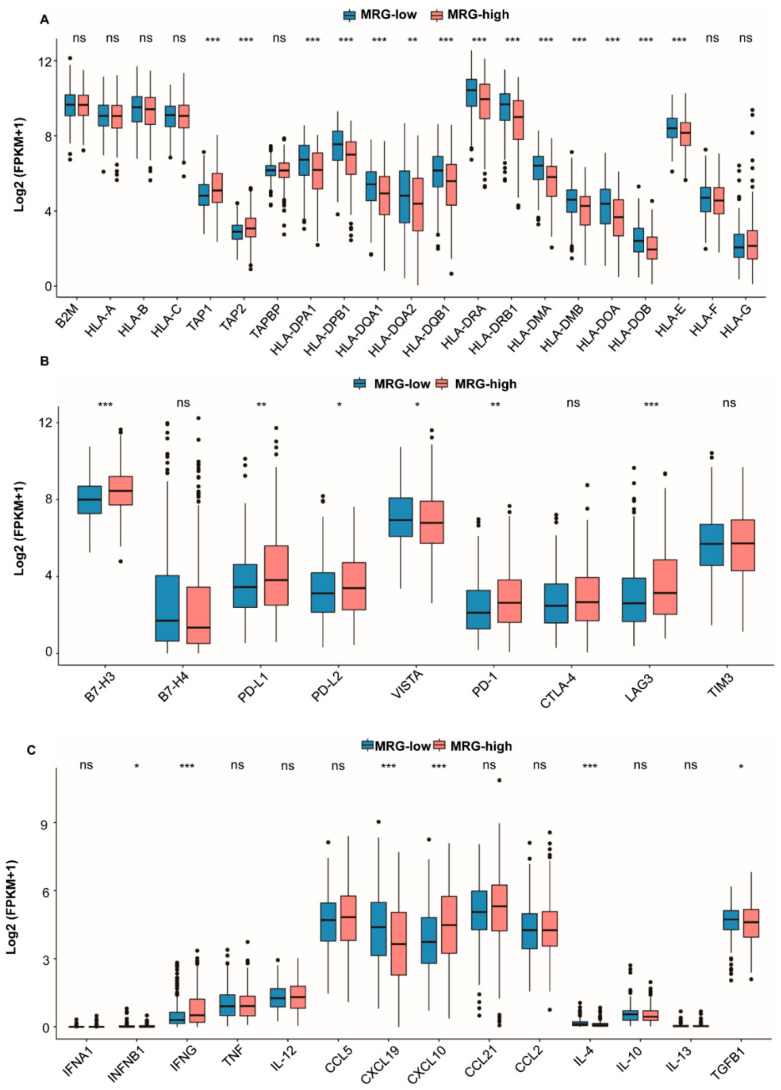
Expression of immunomodulators in different MRG subgroups from TCGA cohort (*n* = 500). (**A**) Major histocompatibility complex (MHC) molecules expression patterns in different MRG subgroups. Significant statistical differences between the two subgroups were assessed using the Wilcoxon test (ns, not significant; **, *p* < 0.01; ***, *p* < 0.001). (**B**) Inhibitory immune checkpoints molecules expression patterns in both MRG subgroups. Significant statistical differences between the two subgroups were assessed using the Wilcoxon test (ns, not significant; * *p* < 0.05; ** *p* < 0.01; *** *p* < 0.001). (**C**) Cytokine expression patterns in both MRG subgroups. Significant statistical differences between the two subgroups were assessed using the Wilcoxon test (ns, not significant; * *p* < 0.05; *** *p* < 0.001).

**Figure 5 ijms-23-12143-f005:**
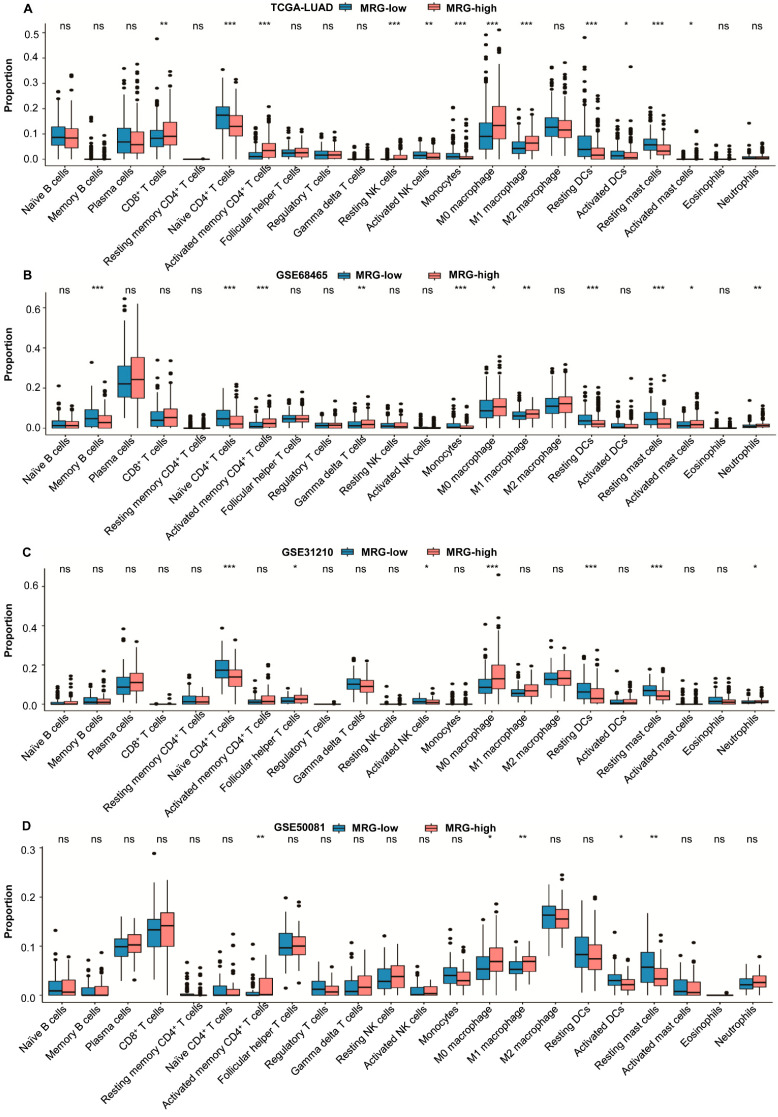
The infiltrated immune cells in different MRG subgroups. (**A**) The proportions of infiltrated cells in different MRG subgroups from TCGA cohort (*n* = 500). (**B**) The proportions of infiltrated cells in different MRG subgroups from GSE68465 cohort (*n* = 442). (**C**) The proportions of infiltrated cells in different MRG subgroups from GSE31210 cohort (*n* = 226). (**D**) The proportions of infiltrated cells in different MRG subgroups from GSE50081 cohort (*n* = 127). The scattered dots represent the immune score. The thick lines represent the median value. The bottom and top of the boxes are the 25th and 75th percentiles (interquartile range), respectively. Significant statistical differences between the two subgroups were assessed using the Wilcoxon test (ns, not significant; * *p* < 0.05; ** *p* < 0.01; *** *p* < 0.001).

**Figure 6 ijms-23-12143-f006:**
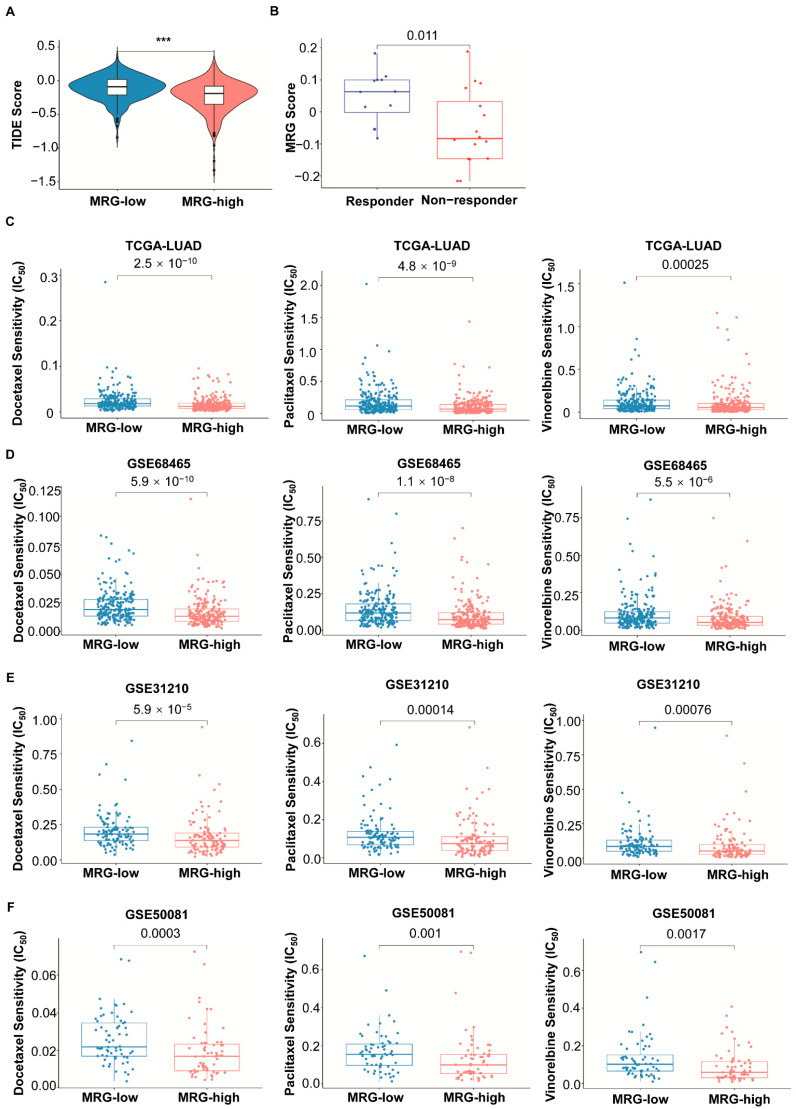
The prognostic value of MRG in patients with different therapies. (**A**) TIDE score in different MRG subgroups. Significant statistical differences between the two subgroups were assessed using the Wilcoxon test (***, *p* < 0.001). (**B**) MRG score of responder and non-responder of ICIs in LUAD patients. (GSE126044 (*n* = 16) and GSE93157 (*n* = 12)). Significant statistical differences between the two subgroups were assessed using the Wilcoxon test. *p* value is shown in the figure. (**C**–**F**) Response to chemotherapeutic agents in both MRG subgroups from TCGA (*n* = 500), GSE68465 (*n* = 442), GSE31210 (*n* = 226), and GSE50081 (*n* = 127), respectively. Significant statistical differences between the two subgroups were assessed using the Wilcoxon test. *p* value is shown in the figures.

## Data Availability

Only publicly available data were used in this study, and data sources and handling of these data are described in the Materials and Methods and in the Appendix A. Further information is available from the corresponding author upon request.

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
