# Peer review of "A Metabolism-Related Gene Prognostic Index for Prediction of Response to Immunotherapy in Lung Adenocarcinoma"

_ijms, 2022, doi:10.3390/ijms232012143_

Round 1

Reviewer 1 Report

I suggest publishing the article in its current form.

I first reviewed the manuscript and then put it in context with the literature. I examined the work of other researchers on the subject, as well as the authors' own workpublications. I checked for possible plagiarism, but found none. Similar work based on database analysis is becoming more and more common. Indeed, it is my opinion that the authors have approached the topic from several points of view and analysed it thoroughly. The introduction and the placement of the results in the literature were appropriate. The literature cited is correct and free from unnecessary self-citation. Of course, any work could be done more perfectly, but I believe that this manuscript is of a high enough standard to be published without change.

Author Response

Thank you so much for your time in reviewing our manuscript and comments. To perfect our work, we have made some modifications throughout the manuscript according to other reviewers. Two more GEO datasets (GSE31210 (n=226) and GSE50081 (n=127)) have been downloaded and processed to verify the accuracy of the MRG model in predicting prognosis. We also downloaded the ICI response data from patients of LUAD (GSE126044 (n=16, LUAD) and GSE31587 (n=12, LUAD)) to validate the MRG model in predicting ICI response.

Reviewer 2 Report

Chuan Xu et al constructed a prognostic model as metabolism-related gene (MRG) of four genes by using weighted gene co-expression network analysis (WGCNA), the nonnegative matrix factorization (NMF), and Cox regression analysis of a LUAD dataset (n=535) from The Cancer Genome Atlas (TCGA), then validated with the Gene Expression Omnibus (GEO) dataset (n=464). The MRG was constructed based on BIRC5, PLK1, CDKN3, and CYP4B1 genes. They found that MRG-high patients had worse survival probability than MRG-low patients. Besides, the MRG-high subgroup was more associated with cell cycle related pathways; high infiltration of CD8+ T cells and M0/1 macrophages; showed better response to ICIs. Contrarily, the MRG-low subgroup was associated with fatty acid metabolism; high infiltration of dendric cells and resting mast cells; showed poor response to ICIs. These results indicate that MRG is a promising prognostic index for predicting survival and response to ICIs and other therapeutic agents in LUAD, which might provide insights on strategies with ICIs alone or combined with other agents. However, the last results show that 4 MRGs also could predict the sensitivity of chemical agents and other targeted drugs, indicating that MRGs are not specific to ICIs. I also doubt that if MRG could be indicator for ICI effects, cause the authors have not explain how MRG could be associated with immunotherapy, such as ICIs.

Other concerns:

1. The authors firstly optimally fractionated 72 genes into two subtypes, C1 and C2 according to the NMF algorithm. Then found that the patients in C2 subtype showed poorer overall survival (OS) and progression-free survival; CD8+ T cells, activated memory CD4+ T cells and M1 macrophages in the patients from C2 subtype were significantly higher than those in the patients from C1 subtype. Finally, the authors obtained four genes (BIRC5, PLK1, CDKN3, 139 and CYP4B1) with a minimal AIC value of 1919.42. Thus, the patients could be grouped as MRG-high subgroup and MRG-low subgroup. So my question is: could MRG-high subgroup can be matched with C2?

2. In the present study, one GEO LUAD dataset (GSE68465, n=462) was used to validate the role of MRG (Fig. S5B). As a study major in informatics analysis, at least two or more datasets should be used to validate the MRG prediction ability.

3. In the study of exploring molecular characteristics between the two MRG subgroups, the authors performed GSEA analysis with annotations of KEGG, metabolic processes, gene mutations, as well as the immune characteristics. Also, at least two or more datasets should be used to validate further, especially immune charateristics.

4. In Results 2.4, the ICI targeted cell or immune checkpoint molecules, or cytokines could be emphasized, and analyzed deeply.

5. I pay attention to that the authors analyzed the sensitivity to some targeted and chemotherapeutic agents. The results showed that patients in the MRG-low subgroup might be more sensitive to Lapatinib, while patients in the MRG-high subgroup might be more sensitive to targeted agents, including Erlotinib and Gefitinib, and chemotherapeutic agents, including Cisplatin, Paclitaxel, Gemcitabine, Etoposide, Docetaxel, and Vinorelbine. So why not to validate these results in more cohorts and widen the use of 4-MRG-genes panel, instead of limit in ICIs.

6. In some of Figures, besides GSE number, the Number of cohort should be indicated, for example, in Fig S10, the number of patients in GSE93157, GSE78220, PRJEB23709,etc has not shown in figure legends.

Author Response

Thank you so much for reviewing our manuscript. The response is attached. Please see the attachment. 

Reviewer 3 Report

In the manuscript „A Metabolism-Related Gene Prognostic Index for Prediction of Response to Immunotherapy in Lung Adenocarcinoma” Tang & Hu and colleagues describe the results of bioinformatic analyses, based on TCGA/GEO datasets derived from LUAD patients. The analyses let the Authors define 4-gene signature, predicting LUAD patient survival time, progression free survival, and correlation with certain aspects of immune cells activation within tumour.

The lack of empirical evidence indicating prognostic value of 4-gene signature (MRG) in immune checkpoint inhibitor therapy is the greatest weakness of the paper. No data from LUAD patient treated/untreated with ICI are analysed to confirm that high MRG score contributes to ICI regimens efficacy. The paper is purely theoretical, and prognostic value of selected gene signature for ICI administration in LUAD patient wasn’t proved convincingly enough to employ MRG signature in the clinics.

There are also other issues: 

1.       The manuscript construction is very chaotic. The figures are divided into supplementary and main groups, but the distribution of results to the figure group is not clear. Especially, the later portion of the story, when other tumour types and cell lines (?) are analysed, loses its flow.

2.       Although some attempts were made to show predictive value of MRG score in other type of cancer, including melanoma, urothelial carcinoma etc. where patients were treated with ICI, this goal wasn’t reached. The presented data fail to convince that MRG score may help in therapeutic decisions in any type of cancer.

3.       It is also unclear, why these 4 differentially genes used to subgroup LUAD tumours are called “metabolism-related”. The group comprises of 2 genes regulating cell cycle, 1 regulating apoptosis, and 1 bona fide metabolic enzyme with monooxygenase activity. MRG-low and MRG-high groups are obviously metabolically different, with MRG-low showing low expression of some genes, mainly related to cell cycle and DNA repair, but high expression of genes belonging to different metabolic pathways, like fatty acid metabolism or drug detoxification etc. (Figure 2 C). Therefore, the name “metabolism-related genes” is not adequate. 

There are some interesting conclusions coming from the analyses provided by Authors, concerning the correlation of 4 gene signature with clinical characteristics of LUAD patients and molecular/immunological properties of LUAD tumour samples. Obviously, high MRG score correlates positively with proliferation rate of the tumour cells (Figure 2A, 2C), TMB (Figure S6) and the expression of some immune checkpoint molecules (Figure S7 B), but these data are not sufficient to validate MRG score as a predictor of ICI therapy response. For instance, PD-L1 protein level is used as a biomarker predicting anti-PD-L1 response, but even lack of PD-L1 protein doesn’t disqualify patients from therapy (https://www.ncbi.nlm.nih.gov/pmc/articles/PMC8750062/). And in the revised manuscript the difference in immune checkpoint molecules gene expression is small.

Author Response

Thank you so much for your time and precious feedback. The response is attached. Please see the attachment. 

Round 2

Reviewer 2 Report

I am satisfied with the revision. It can be accepted at present form.

Author Response

Thank you so much for your time and efforts in reviewing our manuscript. We really appreciate it. 

Reviewer 3 Report

The corrections and rearrangements made to the manuscript have improved its quality significantly, especially due for adding Figure 6B, showing data on response to immunotherapy in LUAD patients. Addition of extra GSEA datasets, as suggested by Reviewer 2, also greatly strengthens the manuscript.

However,  there are still some smaller issues to be clarified:

1.       Please provide the full name of TIDE algorithm, and explain why lower TIDE score predicts better response to immunotherapy.

2.       Please explain what “responder” or “non-responder” to ICI means (Figure 6B). What is the survival time for patients from MRG high and low groups treated with ICIs? The patient number here is low, therefore it could be nice to see patient data in the tabular form (treatment, PFS/OS if ready, MRG score value etc).

3.       Figure 1B – the lowest survival graph probably stands for PLK1, not MRG?

4.       Figure 1 – please clarify whether MRG here comprises 15 or 4 genes or at which point those 15 genes transform to 4 genes.

5.       Figure S4 – CACNA2D2 effect on survival is depicted on two graphs, the second graph represents another gene?

6.       Figure S11 data are not discussed at all in the manuscript, although immunological C1-C6 subgrouping defines MRG-low LUAD group as more inflammatory and MRG-high one as related to wound healing, and these data are worth to be explored further in the discussion.

7.       There are interesting differences in the expression of antigen presenting genes between MRG high and low groups, but these data have never been interpreted/discussed in the manuscript.

8.       There are some problems with positioning of statistical significance symbols along the chart bars, especially in Figure 4A.

9.       Figure S12 –What is a rationale to apply  MRG score established to predict LUAD responses for prediction of melanoma etc. responses to ICI? Is four gene MRG universal for every neoplasm? So maybe these mixed results are not surprising?

10.   Figure  6 C-F – The graphs are generated with OncoPredict algorithm. But how was it done and how to interpret the data? Are MRG-high patients supposed to be more sensitive to chemo drugs? Patients from MRG high group were shown to have worse prognosis (Figure 1) in TCGA and GSEA cohorts.  Most of them were treated with genotoxic drugs, including taxanes? How does it correspond to the greater sensitivity of MRG high group to the selected drugs, shown in Figure 1? 

Author Response

Thank you very much for your time and efforts in reviewing our manuscript. The point-by-point response is attached. Please see the attachment.
